# G-Protein-Coupled Receptor-Associated Sorting Protein 1 Overexpression Is Involved in the Progression of Benign Prostatic Hyperplasia, Early-Stage Prostatic Malignant Diseases, and Prostate Cancer

**DOI:** 10.3390/cancers16213659

**Published:** 2024-10-30

**Authors:** Cesar Torres-Luna, Shuanzeng Wei, Sreenivas Bhattiprolu, George Tuszynski, Vicki L. Rothman, Declan McNulty, Jeff Yang, Frank N. Chang

**Affiliations:** 1Halcyon Diagnostics, 1200 Corporate Blvd. Ste. 10C, Lancaster, PA 17601, USA; cesar.torres@halcyonrx.com (C.T.-L.); george.tuszynski@halcyonrx.com (G.T.); vrothman7@gmail.com (V.L.R.); declan.mcnulty@lynthera.com (D.M.); jeff@halcyonrx.com (J.Y.); 2Department of Pathology, Fox Chase Cancer Center, 333 Cottman Avenue, Philadelphia, PA 19111, USA; weishuanzeng@hotmail.com; 3ZEISS Microscopy Customer Center, 5300 Central Parkway, Dublin, CA 94568, USA; sreenivas.bhattiprolu@zeiss.com

**Keywords:** prostate cancer biomarker, benign prostatic hyperplasia, immunohistochemistry, GASP-1, cancer prevention, ELISA, 2D HPLE, cancer diagnostic

## Abstract

Prostate cancer is the second most common cancer diagnosed in men and yet is the second leading cause of cancer-related deaths worldwide. This is often due to the cancer being misdiagnosed as benign prostatic hyperplasia or missed entirely due to the limited nature of the most popular detection method, prostate-specific antigen levels. The aim of this study was to assess the potential of G-protein-coupled receptor-associated sorting protein 1 (GASP-1) as a valid prostate cancer biomarker. Prostate tissue samples from healthy, benign prostatic hyperplasia, and prostate cancer patients were subjected to anti-GASP-1 polyclonal antibodies in immunohistochemical staining and enzyme-linked immunosorbent assay analyses, showing that GASP-1 levels were significantly greater in prostate cancer patients when compared to benign prostatic hyperplasia and healthy patients. Specifically in prostate cancer patients, there was a positive correlation between GASP-1 overexpression and the severity of the prostate cancer.

## 1. Introduction

Prostate cancer (PCa) is the most frequently diagnosed cancer in men, accounting for more than a quarter of all malignancies found in North American men, and the second cause of cancer-related death [1]. The prediction of PCa is typically based upon serum prostate-specific antigen (PSA) and digital rectal examination results [2]. However, the diagnostic confirmation of prostate cancer in patients with PSA levels in the gray zone (4–10 ng/mL) is controversial, often leading to unnecessary biopsies. Many studies have reported that the traditional PSA cut-off (>4 ng/mL) value is too high, and a significant number of patients with prostate cancer have been reported at a PSA level less than 4 ng/mL [3]. PCa is detected at a rate of about 26% in American men with PSA levels between 2.5 and 4.0 ng/mL and clinically significant cancer was diagnosed in 18.5% of these patients [3,4]. To reduce the over-diagnosis rate and unnecessary treatment of patients with benign conditions, there is a need for more accurate tests to stratify risk in men who present with symptoms of PCa.

PSA is a kallikrein-like serine protease produced by the epithelial cells of the prostate to help liquefy ejaculate and aid sperm motility [5]. This means that PSA is mostly present in semen and only a small amount is in the blood. PSA is also not a highly specific cancer biomarker because serum PSA levels are also abnormally high in benign prostatic hyperplasia (BPH) and other benign conditions including prostatitis and the inflammation of the prostate [6]. BPH is an enlargement of the prostate caused by cellular hyperplasia that occurs within the transition zone [7]. Serum PSA levels are known to increase with age, which is most likely due to the contribution of an enlarged prostate, such as BPH, as well as a decreased retention of the prostatic epithelium. Even though the underlying pathophysiology between BPH and PCa remains unclear, the consensus is that they are separate events—for example, a systematic review that included 16 case–control studies and 10 cohort studies concluded that BPH is associated with an increased risk of prostate cancer although this association was stronger among Asian populations than Caucasians [8]. Aggressive BPH has also been implicated in an elevated risk of developing PCa and the subsequent cancer can be high-grade compared to individuals without fast-growing BPH [2]. The above evidence would support the hypothesis that BPH could be a risk factor involved in the pathogenesis of PCa [2]. Despite these observations, the question remains whether BPH will progress into PCa. With this uncertainty, it would be important to assess the severity of BPH and distinguish BPH patients that may have the potential of developing early-stage PCa. PSA testing alone is not able to make this differentiation. As a result, new biomarkers are required to improve the risk stratification of patients at this stage. Furthermore, a diagnostic assay that could detect PCa at early stages would help to improve treatment options and reduce the risk of cancer aggressiveness or metastasis.

The most common type of PCa is acinar adenocarcinoma, in which cancer cells form glands and cribriform structures that lack basal epithelial cells and are classified using Gleason grading [9]. The Gleason grading, created in 1966, has been adopted for more than 50 years and is one of the most important prognostic factors in men with PCa [10]. Both intraductal carcinoma of the prostate (IDC-P) and ductal adenocarcinoma have been reported to be associated with poor patient outcomes. IDC-P is characterized by prostate carcinoma cells growing within native prostatic ducts and/or acini [11]. These tumors are usually associated with adverse pathological features such as a high Gleason score, large tumor volume, and advanced tumor stage [11]. Ductal adenocarcinoma is recognized by a columnar pseudostratified epithelium with elongated nuclei [11]. Both IDC-P and ductal adenocarcinoma are often under-recognized and even sometimes confused with one another, in both the laboratory and the clinic. In addition, high-grade prostatic intraepithelial neoplasia (HGPIN) is considered a pre-cancerous condition because it can evolve into prostate cancer over time [12]. In fact, some authors consider HGPIN as the most likely precursor of prostatic adenocarcinoma [13].

We have previously identified that G-protein-coupled receptor-associated sorting protein 1 (GASP-1) is a ubiquitous tumor marker that is required for cancer progression and invasion [14,15,16,17,18]. For example, in thyroid cancer, we observed that the expression level of GASP-1 appears to correlate with the metabolic activity of normal thyroid follicular cells and as cancer progresses, more GASP-1 is overexpressed [18]. GASP-1 is minimally expressed in benign thyroid lesions but becomes significantly overexpressed in early-stage malignant thyroid neoplasms [18]. In breast cancer, we also showed that GASP-1 promotes the proliferation and invasion of the triple-negative breast cancer cell line MDA-MB-231 [17]. Furthermore, it was demonstrated that a polyclonal antibody against a specific GASP-1 peptide EEASPEAVAGVGFESK inhibited growth and reduced the size of breast cancer cell colonies in soft agar by more than 90% [15,16,17]. This peptide sequence was chosen because it was detected from serum albumin complexes using our proprietary two-dimensional high-performance liquid electrophoresis (2-D HPLE) procedure that separates cancer protein complexes using a polyvinylidene difluoride (PVDF) membrane [19]. GASP-1 is a large protein with 1394 amino acids that belongs to the family that regulates G-protein-coupled receptors (GPCRs), which represent one of the most abundant receptor networks encoded by nearly 4% of the human genome. GASP-1 is likely to have many fragments circulating in the blood. Focusing on a specific GASP-1 peptide fragment known to be in the circulation would increase the sensitivity of detection. To the best of our knowledge, GASP-1 has not been reported as a biomarker in the development and progression of both BPH and PCa. For this paper, we developed a GASP-1 ELISA that can distinguish between BPH and PCa using GASP-1 serum levels, which vary between the two by a 5-fold difference. The ELISA result was confirmed by conducting GASP-1 immunohistochemistry. Our results show that GASP-1 overexpression is involved in the progression of BPH, early-stage malignant diseases, and prostate cancer.

## 2. Materials and Methods

### 2.1. Antibody Production

Anti-GASP-1 polyclonal antibodies against EEASPEAVAGVGFESK were produced by ABclonal Technology, Inc. (Woburn, MA, USA). Affinity-purified IgG on a column of the GASP-1 peptide EEASPEAVAGVGFESK was used for routine IHC and ELISA analyses.

### 2.2. Prostate Tissue Microarray

Prostate tissue microarray PR807c was purchased from USBiomax, Inc (Derwood, MD, USA) and used for GASP-1 immunohistochemistry (IHC). The prostate cancer microarray contained the Gleason score information assessed by their board-certified pathologist.

### 2.3. Tissue Staining

Anti-GASP-1 polyclonal antibodies were used in IHC staining to detect GASP-1 in formalin-fixed, paraffin-embedded (FFPE) tumor tissues as reported previously [14]. In brief, tissue staining was performed by Discovery Life Sciences (Newtown, PA, USA) according to standard procedures using optimized antibody concentration and antigen retrieval and the appropriate negative controls. Isotype non-immune IgG and immune sera adsorbed with a GASP-1 peptide were used as controls and showed no staining. For GASP-1 IHC, we analyzed 3 healthy individuals, 20 patients with BPH, and 33 patients with prostate cancer.

### 2.4. GASP-1 IHC Scoring

We reported previously that the overexpression of GASP-1 is required for the progression of many cancers [15,16,17,18]. Depending on the severity of cancer, the overexpressed GASP-1 in the cytoplasm of cancer cells aggregates to form granules of various sizes including powdery, fine, and coarse granules [14]. To quantify the GASP-1 granules from BPH and various stages of prostate cancers, a new IHC scoring system (called the “H-score”) was developed by Dr. Richard Siderits at Discovery Life Sciences (Newtown, PA, USA) [14]. Briefly, the GASP-1 granule quantification involves recording the percentage of cells (tumor or normal) with GASP-1 cytoplasmic staining at a corresponding differential intensity on a four-point scale semi-quantitatively (0, 1+, 2+, 3+). The H-score is calculated by summing the percentage of cells with the intensity of expression (brown staining) multiplied by their corresponding differential intensity on a scale from 0 to 3+. H-scores range from 0 to 300. A more detailed description of the H-score can be found in our previous study [14]. The Gleason score was determined according to the literature [20,21].

### 2.5. GASP-1 ELISA

The ELISA procedure involves a conjugate of BSA (bovine serum albumin) and a GASP-1 peptide (BSA-GASP-1 conjugate) to coat a single plate that retains the GASP-1 peptide before detecting GASP-1 in a serum sample in a competitive assay. The sequence of the GASP-1 peptide is EEASPEAVAGVGFESK. The details of the procedure were reported previously [14]. The normal and BPH samples were obtained from Discovery Life Sciences (Los Osos, CA, USA). Prostate cancer serum samples were obtained from Dr. Leland Chung of Cedars-Sinai Medical Center (Los Angeles, CA, USA). Samples were obtained from patients that were diagnosed with advanced metastatic prostate cancer (Gleason Grade 4 or 5) at the time samples were obtained. All patient samples were collected with informed consent from the donors and their relatives. We analyzed serum samples from 5 healthy individuals, 11 patients with BPH, and 17 patients with prostate cancer.

### 2.6. Statistical Analysis

For the analysis of ELISA results, we performed one-way ANOVA by using the stats package from the scipy library in Python (version number 1.11.3). It provides a function called ‘f_oneway’ that calculates the F-statistic and *p* value (https://docs.scipy.org/doc/scipy/reference/generated/scipy.stats.f_oneway.html (accessed on 23 October 2024)). GraphPad Prism (version 8.0.0) was also used to perform statistical analyses. The statistical significant difference for two sample comparisons was determined by an unpaired Student *t* test. A *p* value of <0.05 was considered statistically significant.

### 2.7. Image Analysis: Tissue Microarray (TMA) De-Arraying

To analyze individual cores, the TMA was subjected to de-arraying using QuPath software (Version 0.4.0) and a custom Python code [22]. QuPath facilitated loading the TMA, extracting core locations, and saving them to a text file. Subsequently, the custom Python code processed the core positions, extracting and saving individual cores as PNG files at a native resolution (Level = 0) for downstream digital stain separation.

### 2.8. Digital Stain Separation

For precise assessment of GASP-1 expression, blue and brown stains were digitally separated within each core. This process followed the methodology outlined by Ruifrok et al., utilizing the hematoxylin–eosin–DAB (HED) color space conversion in the scikit-image library′s Python method [23]. The resulting images were saved as PNG files for a subsequent analysis [24].

## 3. Results

### 3.1. Differentiating BPH from Prostate Cancer via GASP-1 ELISA

For decades, the PSA test has been used for detecting prostate cancer. As indicated earlier, GASP-1 overexpression is required for cancer progression and invasion, and as cancer progresses, more GASP-1 is produced and released into the circulation [15,16,17]. Therefore, when compared to prostate cancer, we would expect BPH to have lower GASP-1 expression levels. Figure 1 shows that in our GASP-1 ELISA, there is a 5-fold difference in serum GASP-1 levels between BPH and prostate cancer patients. A 20-fold difference in serum GASP-1 levels between healthy individuals and prostate cancer patients was also found. To confirm the statistical significance of our results, we performed a one-way ANOVA test, and it yielded an F-statistic of 4.5038 and a *p* value of 0.01948. Since the *p* value is less than 0.05, we conclude that there is a statistically significant difference in GASP-1 levels between the healthy, BPH, and prostate cancer groups. Based on these results, our GASP-1 ELISA can differentiate between BPH and prostate cancer and could potentially be used to supplement the PSA test.

### 3.2. Confirmation of ELISA Results via GASP-1 Immunohistochemistry

To verify the ELISA results presented above, we assessed the level of GASP-1 expression from normal tissue samples, BPH tissue samples, and tissue samples of different grades of prostate cancer via immunohistochemical staining with hematoxylin and GASP-1. As indicated earlier, when GASP-1 is overexpressed in cancers, it starts to aggregate and form granules of various sizes depending upon the severity of cancer [14]. We have developed a new H-scoring system that captures and quantifies the novel features of GASP-1 expression including the formation of granules. Even though traditional scoring systems may reflect important features of GASP-1 expression, they fail to capture cancer prognostic significance. Figure 2 shows that GASP-1 is very minimally expressed in normal cells and has very low H-scores. The progression of normal cells to BPH leads to the production of more GASP-1 and an increase in the H-score, reaching an average value of 65.7 for the 20 BPH samples. In prostate cancer, much more GASP-1 was expressed, resulting in an increase in the H-score to an average value of 180.5, or about three times the value of BPH. A clear separation between BPH and prostate cancer was observed with a *p* value of <0.0001 (compare columns 2 and 3 in Figure 2). Therefore, the ELISA results reported in Figure 1 were confirmed by using GASP-1 IHC.

Because the prostate cancer samples analyzed in Figure 2 contain cancers with different Gleason grades (3, 4, and 5), we decided to separately quantify their GASP-1 expression levels. It is interesting to note that the composite prostate cancer H-scores in column 3 of Figure 2 can be roughly divided into two subgroups. The lower half of prostate cancer samples have an H-score from 100 to 150 and they are almost exclusively coming from Gleason Grade 3 prostate tissue samples (compare columns 3 with 4 in Figure 2). The upper half of the cancer samples have an H-score from around 180 to 260 and they are predominantly from Gleason Grades 4 and 5 (compare columns 3 with 5 in Figure 2). The fact that Gleason Grade 3 prostate cancers have lower GASP-1 H-scores than Gleason Grade 4 and 5 cancers indicates that GASP-1 expression can be used to assess the progression of prostate cancer.

Similarly, the BPH group also exhibited a wide variation in their H-score numbers, which range from a low of 20 to a high of 100 (see column 2 of Figure 2). As a result, BPH samples can also be divided into two subgroups based on their GASP-1 expression levels: low GASP-1 expression (with H-numbers at 60 or below), and high GASP-1 expression (with H-numbers above 60). Since GASP-1 expression is correlated with cell growth [18], it is proposed that BPH with high H-scores is a faster grower, and it could be associated with more aggressive BPH and therefore with an elevated risk of developing into prostate cancer [2]. If our results are verified in a larger population study, this information could be highly beneficial for patients in understanding their status of BPH.

### 3.3. GASP-1 Expression in Normal and BPH Tissues

GASP-1 is synthesized on the endoplasmic reticulum, which is attached to the nucleus and is subsequently released into the cytosol. Figure 3 shows the GASP-1 IHC result from both normal and different BPH tissues. We found that GASP-1 was minimally expressed in normal cells (see Figure 3A). Our result agrees with the report that normal prostatic glandular cells have an extremely low rate of cell proliferation [25]. Figure 3B shows GASP-1 expression from BPH having a low H-score (score of 20). The GASP-1 expression level is very low and almost identical to that of normal cells. BPH cases with a high H-score (score of 100) are shown in Figure 3C,D. Moderate GASP-1 expression was found in both inner epithelial cells (Figure 3C) and basal layer cells (Figure 3D). Our results confirm that BPH cases with higher H-score numbers are more actively growing and could be at a more advanced BPH stage. Interestingly, we also noticed the presence of mini glands, which appear to be released from bigger glands (see arrows in Figure 3D). The function of these mini glands is unknown. However, because they are enriched in GASP-1, they may promote BPH growth.

### 3.4. GASP-1 Granules Can Be Used to Differentiate Benign Condition from Early-Stage Prostate Cancer

As cancer progresses, the overexpressed GASP-1 in the cytoplasm of cancer cells forms granules of various sizes including powdery, fine, and coarse granules [14]. The GASP-1 granules begin to attach to the plasma membranes, which is an abnormal condition that is associated with cancer development [14]. When we compared GASP-1 expression in BPH with a high H-score (Figure 4A) and Gleason Score 3 + 3 PCa (Figure 4B–D), we found that more GASP-1 is synthesized in PCa and its powdery granules begin to attach to the plasma membranes. We also examined the GASP-1 expression of intraductal carcinoma of the prostate (IDC-P) (Figure 4C). The GASP-1 expression profile in IDC-P appears to be similar to that of adenocarcinoma, also shown in Figure 4C. Based on these findings, we conclude that both the size of GASP-1 granules and their attachment to plasma membranes could be used to differentiate a benign condition from early-stage prostate cancer.

### 3.5. GASP-1 IHC Is Superior to Conventional H&E Stain in Identifying Prostate Cancer

Hematoxylin and eosin (H&E) staining has been the most widely used staining method in immunohistochemical diagnoses and is regarded as the gold standard method when a pathologist looks at a biopsy suspected of cancer. By using H&E, one can differentiate between the nuclear and cytoplasmic parts of a cell and the general layout and distribution of cells. Unlike eosin, which stains many proteins, a more targeted approach would use specific biomarkers such as GASP-1 that are required for cell growth to assess the early stage of cancer progression. GASP-1 IHC of a Gleason Score 3 + 3 tissue shows the presence of granules that are attached to the plasma membranes of inner layer cells of prostate ducts (Figure 5B), which are not apparent in the H&E image (Figure 5A). Many GASP-1 granules are also found to be attached to the outer layer cells.

### 3.6. Continuous Overexpression of GASP-1 in Advanced Stages of Prostate Cancer

Figure 6 shows that as prostate cancer progresses, the overexpression of GASP-1 leads to the production of coarse granules with more intense cytoplasmic staining. As expected, powdery GASP-1 granules are attached to the plasma membrane in Gleason 3 + 3 (Figure 6A). The continuous overexpression of GASP-1 leads to the further aggregation of GASP-1 and attachment of coarse-size granules to the plasma membranes as seen in Gleason 4 + 4 (Figure 6B) and Gleason 4 + 5 (Figure 6C). Based on the above observations, the size of GASP-1 granules and their attachment to plasma membranes can also be used as indicators for assessing the progression of prostate cancer. Interestingly, the size of the glands becomes smaller (mini glands) as cancer progresses. Some of the mini glands in Gleason 4 + 5 contain only a few cancer cells (Figure 6C). The GASP-1-enriched mini glands could be important in cancer progression.

### 3.7. Color Segregation for Analysis of Progression of GASP-1 Granules

The synthesis of GASP-1 on the endoplasmic reticulum complicates an IHC image analysis because the nucleus will initially appear as a mixed color or brownish if large amounts of GASP-1 are located there. An open-source software system was developed to segregate the two colors to allow assessment of GASP-1 expression without the interference of the blue nuclear stain. Our color segregation process can be conducted within seconds. Figure 7A–C show the results of regular GASP-1 IHC from different stages of prostate cancer, while Figure 7D–F show only GASP-1 expression without the interference of blue staining. The size and location of GASP-1 granules can be easily visualized without interference. GASP-1 granules are prevalent in Gleason Score 3 + 4 and become highly abundant in Gleason Score 4 + 5. We conclude that as prostate cancer becomes more advanced, more GASP-1 is overexpressed. Therefore, the selective expression of GASP-1 in cancer cells distinguishes it from BPH.

## 4. Discussion

The PSA blood test is mainly used to screen for prostate cancer (PCa) in men with or without symptoms. While PSA testing has helped identify many patients with PCa, a key remaining obstacle for clinicians is the differentiation of PCa from non-malignant conditions. A PSA score of >4 ng/mL has a sensitivity of 93%, but only 20% specificity in PCa detection [26]. Such low specificity makes the test of little use in PCa screening. Only 1 in 4 men with elevated PSA will be diagnosed with PCa, while patients with PSA <1 ng/mL are still 10% likely to develop the disease [27]. Despite the use of several diagnostic models containing clinical data such as patient age, family history of PCa, and PSA derivatives (e.g., PSA density, PSA velocity), PSA as a screening method leads to an over-diagnosis and, consequently, to over-treatment [28]. Even with relatively new biological markers available, such as the TMPRSS2-ERG fusion gene, non-coding RNA (PCA3) [29], kallikrein included in basic PHI (prostate health index), or 4K tests [30,31,32], there is an urgent need for more accurate methods to risk-stratify men who present with symptoms of PCa in order to prevent the over-diagnosis and unnecessary treatment of patients with benign conditions. We believe that our GASP-1 ELISA addresses the deficiencies inherent in the current PSA test and it could represent a better alternative in differentiating BPH from prostate cancer. Our GASP-1 ELISA shows a 5-fold difference in serum GASP-1 levels between BPH and prostate cancer patients. Once verified in larger patient population clinical trials, we would recommend that only individuals showing high levels of both PSA and GASP-1 undergo prostate biopsies. Our ELISA result is also confirmed by GASP-1 immunohistochemical staining, which differentiates BPH from prostate cancer with a *p* value of <0.0001 (see Figure 2).

The potential of BPH to develop into prostate cancer remains an intensely debated issue in the scientific and clinical communities. In this regard, it is interesting to note that we divided BPH into either low or high expression based on GASP-1 levels. While low-expression BPH has a GASP-1 level comparable to that of normal cells, powdery size GASP-1 granules are noticeable in some cases of high-expression BPH (see Figure 3 and Figure 4). Even so, their expression level has not reached the extent exhibited for early-stage prostate cancer. Based on these observations, we could conclude that BPH is benign and does not develop into early-stage prostate cancer. However, we could not rule out the possibility that if GASP-1 continues to be expressed in fast-growing BPH and form bigger aggregates that are attached to plasma membranes, then these fast-growing BPH cases could develop into early-stage cancer. Importantly, our results indicate that the overexpression of GASP-1 contributes to the development of prostate cancer. Other factors have been shown to have prognostic significance in prostate cancer. For example, thyroid hormones are known to affect prostate cancer [33]. Hyperthyroidism apparently increases the risk of both BPH and prostate cancer [33]. A higher expression of TGF-β was reported in tumor tissues with a higher Gleason score [34]. Thus, these factors along with GASP-1 could have a very strong prognostic significance.

Compared to conventional H&E staining, our GASP-1 IHC is more sensitive in detecting prostate cancer. The ability to detect early-stage prostate cancer and assess its severity will be highly beneficial to prostate cancer patients because assessment based on H&E could miss early-stage prostate cancer. Using our H-score analysis, GASP-1 IHC can assess the severity of early prostate cancer. In our limited-sample-size analysis, GASP-1 IHC could be used to differentiate early and more advanced prostate cancer (such as between Gleason Grades 3 and Gleason Grades 4 and 5) with a *p* value of 0.0004 (see columns 4 and 5 of Figure 2). However, analyzing a larger number of patient samples would be required to validate the obtained results.

To facilitate the color analysis of GASP-1 granules, we have developed an open-source software system, which can segregate within seconds the blue hematoxylin color from the brown staining in our GASP-1 IHC. The size of GASP-1 granules can then be easily assessed during the progression of prostate cancer. Our software system would also be useful for the analysis of the expression of proteins that are synthesized on the endoplasmic reticulum and destined to be exported without the complication of the nuclear hematoxylin staining.

While the study highlights the promising potential of GASP-1 as a biomarker for prostate cancer, it also has notable limitations. The patient cohort was small, with a limited number of healthy individuals in the control groups in the GASP-1 ELISA and IHC scoring. Therefore, further validation in a larger and more diverse cohort of patient samples is essential to confirm the findings and enhance the robustness of the conclusions. Given that prostate cancer predominantly affects older men, recruiting a sufficient number of eligible participants within this demographic can be challenging, particularly when the study specifies certain disease stages. Additionally, the absence of data on PSA levels for comparison between H-scoring and GASP-1 serum levels hinders the ability to establish a correlation between GASP-1 expression and PSA levels.

## 5. Conclusions

This study introduces G-protein-coupled receptor-associated sorting protein 1 (GASP-1) as a potential biomarker for the detection of prostate cancer (PCa), aiming to address the limitations of current diagnostic tools, primarily serum prostate-specific antigen (PSA) levels. The current study explores the utility of GASP-1 through immunohistochemistry (IHC) and enzyme-linked immunosorbent assay (ELISA) analyses, revealing a 5-fold difference in serum GASP-1 levels between benign prostatic hyperplasia (BPH) and PCa patients. The novel scoring system, the H-score, assesses GASP-1 granules′ intensity and size, providing a clear distinction between BPH and PCa. Notably, GASP-1 overexpression correlates with PCa severity, offering insights into disease progression. The study suggests GASP-1 as a promising diagnostic marker that could supplement PSA testing and improve risk stratification for PCa patients. Additionally, an open-source software system is introduced for an efficient GASP-1 granule color analysis, enhancing diagnostic accuracy. The potential of GASP-1 in differentiating BPH from PCa and assessing disease progression makes it a valuable candidate for further exploration in clinical settings.

## Figures and Tables

**Figure 1 cancers-16-03659-f001:**
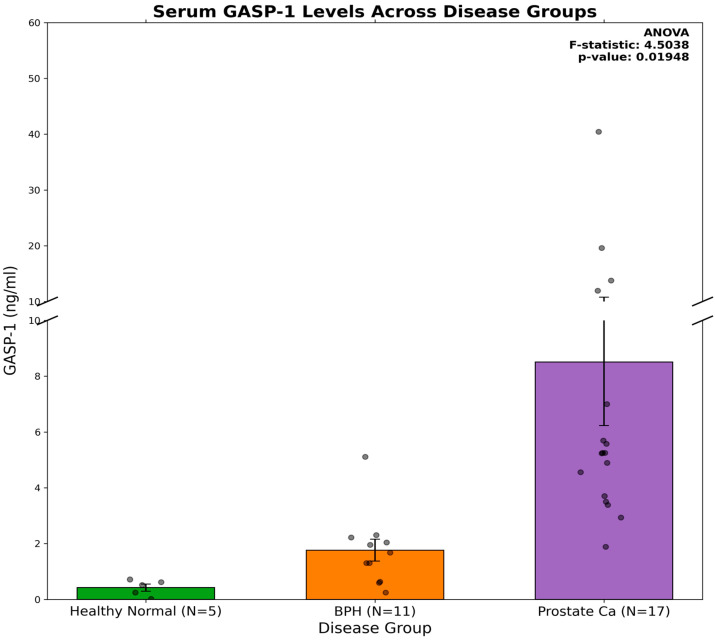
GASP-1 ELISA for differentiating BPH from prostate cancer.

**Figure 2 cancers-16-03659-f002:**
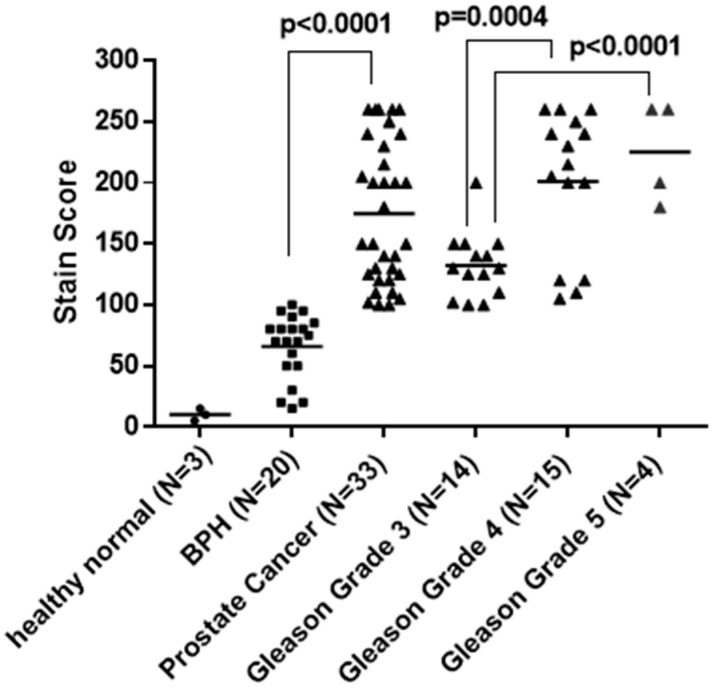
Overexpression of GASP-1 in BPH and different Gleason grades of prostate cancer. Staining score of GASP-1 expression was achieved by immunohistochemistry.

**Figure 3 cancers-16-03659-f003:**
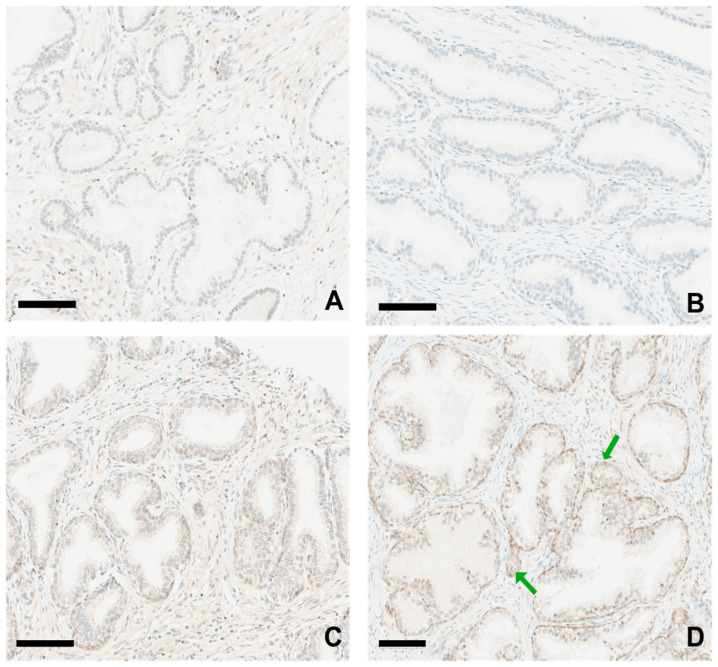
GASP-1 expression in normal and different BPH tissues. (**A**) Normal prostate tissue. (**B**) BPH case with low H-score (H = 20); (**C**,**D**) BPH cases with high H-score (H = 100). Scale bar = 100 µm. Green arrows in (**D**) point out mini glands.

**Figure 4 cancers-16-03659-f004:**
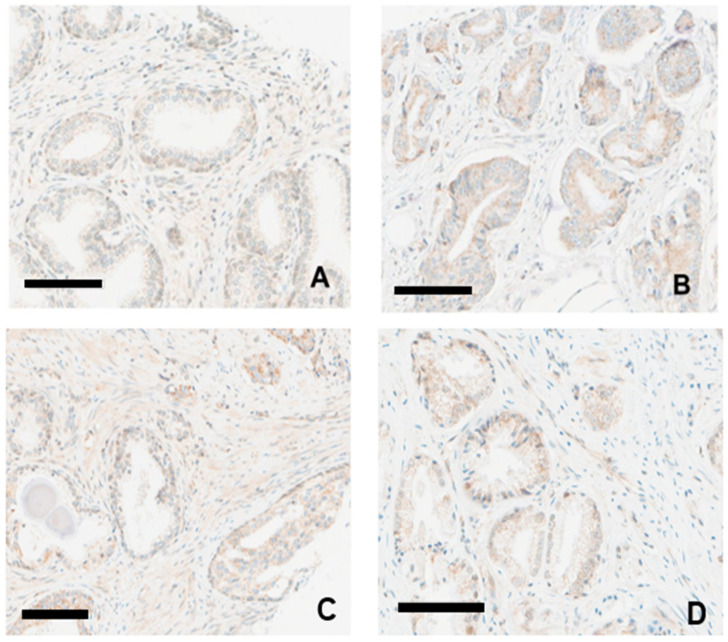
GASP-1 expression in BPH and in prostate cancer. (**A**) BPH case with high H-score (H = 100); (**B**) prostate cancer with Gleason Score 3 + 3 (H = 120); (**C**) intraductal carcinoma of prostate (IDC-P) in lower right corner (H = 110), with adenocarcinoma in upper right corner; and (**D**) prostate cancer with Gleason Score 3 + 3 (H = 110). Scale bar = 100 µm.

**Figure 5 cancers-16-03659-f005:**
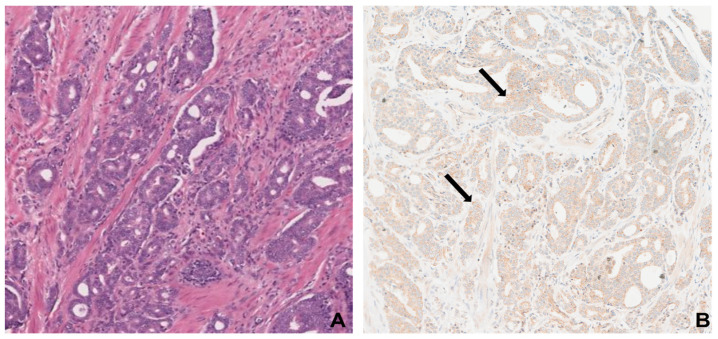
Comparison between H&E (**A**) and GASP-1 IHC (**B**) in same prostate cancer tissue (Gleason Score 3 + 3). In (**B**), black arrows point out GASP-1 granules.

**Figure 6 cancers-16-03659-f006:**
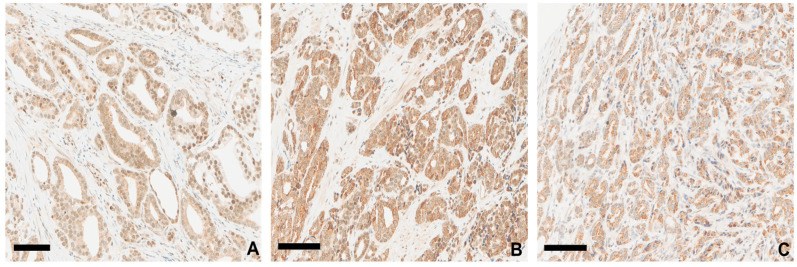
GASP-1 overexpression in different prostate cancer cases. (**A**) Gleason 3 + 3; (**B**) Gleason 4 + 4; (**C**) Gleason 4 + 5. Scale bar = 100 µm.

**Figure 7 cancers-16-03659-f007:**
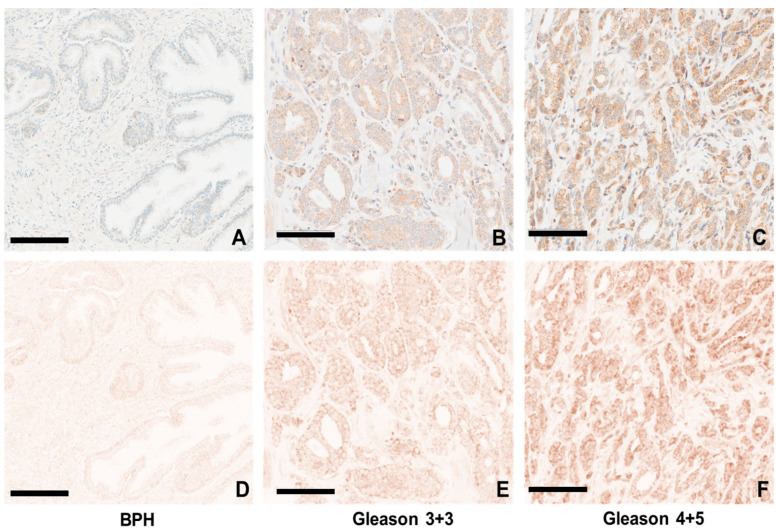
Analyses of GASP-1 expression in various stages of prostate cancer after color segregation. (**A**–**C**) are regular GASP-1 IHC while (**D**–**F**) show only GASP-1 expression without the interference of blue staining. (**A**,**D**) are for a BPH case; (**B**,**E**) are for a Gleason 3 + 4 case; and (**C**,**F**) are for a Gleason 4 + 5 case. Scale bar = 100 µm.

## Data Availability

The data generated in this study are available upon request from the corresponding author. All of the original code used for the image analysis was deposited at GitHub and is publicly available under the MIT license as of the date of publication. The link is https://github.com/bnsreenu/TMA-dearray-stain-separation (Published on 6 December 2023 and last updated 7 February 2024).

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
