# Peer review of "G-Protein-Coupled Receptor-Associated Sorting Protein 1 Overexpression Is Involved in the Progression of Benign Prostatic Hyperplasia, Early-Stage Prostatic Malignant Diseases, and Prostate Cancer"

_cancers, 2024, doi:10.3390/cancers16213659_

Round 1

Reviewer 1 Report

Comments and Suggestions for Authors

In the article of C. Torres-Luna et al the significant difference in serum GASP-1 levels between benign prostatic hyperplasia (BPH) and prostate cancer (PC) patients was deminstrated. The novel scoring system, assesses GASP-1 granules intensity and size, providing a clear distinction between BPH and PCa.

The main point of the study is to find a new biomarker that can detect prostate cancer at early stages and differentiate it from benign prostatic hyperplasia. The commonly used prostate membrane antigen (PMA) test is not always reliable and often gives false-positive results. GASP-1 is known to be minimally expressed in benign thyroid and breast lesions, but is significantly overexpressed even in early stages of malignant neoplasms. The originality of the work is that GASP-1 is proposed to be used to diagnose early stages of malignant prostate diseases. To date, I have not been able to find similar works by other research groups in the available literature.

The methodological work was performed correctly. Microarrays of prostate tissue from normal patients, patients with BPH and prostate cancer with different Gleason scores were studied, immunohistochemical and enzyme immunoassay were performed using polyclonal antibodies directed against GASP-1. In addition, a new H-score system was used to study the intensity and size of GASP-1 granules, revealing a clear distinction between malignant and benign prostate tumors.

Increased GASP-1 expression correlated with the severity of prostate cancer. The study confirms the role of GASP-1 as a promising diagnostic marker offering improved risk stratification of prostate cancer.

Conclusions are consistent with the data obtained. Available references correspond to the topic of the work.

I believe that the presented data is sufficiently novel and the article may be interesting from both theoretical and practical points of view and can be recommended for publication in the presented form.

Author Response

We thank Reviewer 1 for the time to review our manuscript and for the positive feedback provided. No comments to address.

Reviewer 2 Report

Comments and Suggestions for Authors

Dear Editors,

The manuscript entitled: "GASP-1 overexpression is involved in the progression of BPH, early-stage prostatic malignant diseases, and prostate cancer" is generally a well-written original article investigating the role of GASP-1 as a potential biomarker distinguishing prostate cancer from benign conditions, such as BPH. The study supports GASP-1's role as a promising diagnostic marker, measured both in serum, but also during IHC validation, supplementing PSA testing, and offering improved risk stratification for prostate cancer. 

I have a few minor, but critically important concerns (listed below), which I would like the authors to address.

If the authors are able to address them properly, I recommend that the paper is submitted for another review, before final acceptance.

Minor concerns:

Lines 318-324:

The authors should rather state that: "The PSA blood test is used to screen for prostate cancer (PCa) in men with or without symptoms."

The sentence: "A PSA score of >4 ng/ml has a specificity of 94%, but only 20% sensitivity in PCa detection [26]." contains a mistake, it should be corrected to: 

"A PSA score of >4 ng/ml has a sensitivity of 93%, but only 20% specificity in PCa detection".

I also suggest to cite the following paper [Merriel et al. 2022, https://doi.org/10.1186/s12916-021-02230-y], instead of Porzycki et al. 2020, which refers to Hayes et al. 2014, who doesn't provide appropriate information.

Hence, the sentence: "Such low sensitivity makes the test of little use in PCa screening." 

should sound: "Such low specificity makes the test of little use in PCa screening."

The information given in the sentence:

"Only 1 in 4 men with elevated PSA will be diagnosed with PCa, while patients with PSA<1ng/ml are still 10% likely to develop the disease [27]" does not come from the paper by Hayes et al. 2014, please give the correct citation. 

Line 350: 

is "Importantly, our results indicate that overexpression of GASP-1 expression contributes to the development of prostate cancer."

should be: "Importantly, our results indicate that overexpression of GASP-1 contributes to the development of prostate cancer."

Please, give one separate paragraph at the end of the Discussion summarizing the study limitations.

The 'Institutional Review Board Statement' should be completed. 

Author Response

We thank Reviewer 2 for the time to review our manuscript and for the positive feedback provided. Below are the comments from Reviewer 2 as well as our responses to those.

Minor concerns:

1. Lines 318-324: The authors should rather state that: "The PSA blood test is used to screen for prostate cancer (PCa) in men with or without symptoms."

Response: This was corrected in the revised manuscript

2. The sentence: "A PSA score of >4 ng/ml has a specificity of 94%, but only 20% sensitivity in PCa detection [26]." contains a mistake, it should be corrected to:  "A PSA score of >4 ng/ml has a sensitivity of 93%, but only 20% specificity in PCa detection".

Response: This was corrected in the revised manuscript

3. I also suggest to cite the following paper [Merriel et al. 2022, https://doi.org/10.1186/s12916-021-02230-y], instead of Porzycki et al. 2020, which refers to Hayes et al. 2014, who doesn't provide appropriate information. 

Response: We have cited Merriel et al. 2022 manuscript instead of Porzycki’s manuscript.

4. Hence, the sentence: "Such low sensitivity makes the test of little use in PCa screening."  should sound: "Such low specificity makes the test of little use in PCa screening."

Response: This was corrected in the revised manuscript

5. The information given in the sentence: "Only 1 in 4 men with elevated PSA will be diagnosed with PCa, while patients with PSA<1ng/ml are still 10% likely to develop the disease [27]" does not come from the paper by Hayes et al. 2014, please give the correct citation. 

Response: This was corrected in the revised manuscript, and we gave the correct citation

6. Line 350: is "Importantly, our results indicate that overexpression of GASP-1 expression contributes to the development of prostate cancer."  But should be: "Importantly, our results indicate that overexpression of GASP-1 contributes to the development of prostate cancer."

Response: This was corrected in the revised manuscript

7. Please, give one separate paragraph at the end of the Discussion summarizing the study limitations.

Response: We have added a separate paragraph at the end of the Discussion summarizing the study limitations

8. The 'Institutional Review Board Statement' should be completed. 

Response: Institutional Review Board Statement was added at the end of the manuscript before References

Reviewer 3 Report

Comments and Suggestions for Authors

The presented manuscript by Torres-Luna et al investigated the role of G-protein coupled receptor-associated sorting protein 1 (GASP-1) as a potential biomarker for prostate cancer (PCa) and benign prostatic hyperplasia (BPH). Their study revealed that GASP-1 levels were significantly higher in prostate cancer patients compared to those with BPH and normal prostate tissue, demonstrating a five-fold difference in serum GASP-1 levels. Additionally, they developed a novel H-score scoring system to quantify GASP-1 expression, showing that its overexpression correlated with the severity of prostate cancer, particularly among different Gleason grades. These findings suggest that GASP-1 could serve as a valuable diagnostic marker to improve risk stratification and complement existing PSA testing.

Comments and Suggestions for Authors

1) In the material and methods section, please indicate the number of patients comprising the cohort represented on the tissue microarray. Similarly, it is advisable to indicate the number of serum samples analyzed by ELISA, categorized by pathology type (normal, BPH, and cancer), even though each data point is visible in the results graph.

 2) Regarding the statistical analysis, the Student-T test is not appropriate in this case. You wish to determine whether there are statistically significant differences between several samples as illustrated in your Figure 1 (3 groups). An ANOVA or non-parametric Kruskall-Wallis test is therefore required if the data do not follow a normal distribution (which seems to be the case given the distribution of the Cancer samples). To compare groups 2 by 2, a multiple comparison test (such as Tukey or Dunnett) must then be performed. The statistical analyses therefore need to be revised, and the graphs redone based on the new results.

 3) The same applies to the statistical analyses in Figure 2. In addition, it will be easier to understand if Figure 2 is divided into 2 separate figures: one showing healthy samples, BPH and cancers, and the second comparing the 3 Gleason grades.

 4) Can the authors provide better quality, higher contrast images for figures 3 and 4? The images are pale with a weak signal. In fact, even when zoomed in, the labeling cannot be clearly distinguished. It would be interesting to add an insert showing the image at higher magnification for each photo (3A 3B 3C 3D and 4A 4B 4C 4D).

 5) There is still a long way to go to consider the H-score of GASP-1 applicable in routine. This is why a dedicated section on the limitations of the study would be beneficial in the discussion or conclusion. This must include the need for further validation in larger cohorts.

Author Response

We thank Reviewer 3 for the time to review our manuscript and for the positive feedback provided. Below are the comments from Reviewer 3 as well as our responses to those.

Comments and Suggestions for Authors

1) In the material and methods section, please indicate the number of patients comprising the cohort represented on the tissue microarray. Similarly, it is advisable to indicate the number of serum samples analyzed by ELISA, categorized by pathology type (normal, BPH, and cancer), even though each data point is visible in the results graph.

Response: We have added in the Material and Methods section the number of patients comprising the cohort represented on the tissue microarray as well as the number of serum samples analyzed by ELISA. For GASP-1 IHC, we analyzed 3 normal individuals, 20 patients with BPH and 33 patients with prostate cancer. For ELISA, we analyzed serum samples from 5 normal individuals, 11 patients with BPH and 17 patients with prostate cancer.

 2) Regarding the statistical analysis, the Student-T test is not appropriate in this case. You wish to determine whether there are statistically significant differences between several samples as illustrated in your Figure 1 (3 groups). An ANOVA or non-parametric Kruskall-Wallis test is therefore required if the data do not follow a normal distribution (which seems to be the case given the distribution of the Cancer samples). To compare groups 2 by 2, a multiple comparison test (such as Tukey or Dunnett) must then be performed. The statistical analyses therefore need to be revised, and the graphs redone based on the new results.

Response: We have generated a new Figure 1 using one-way ANOVA for statistical analysis, and we have added sentences in Section 2.6 (Statistical Analysis) describing how we conducted one-way ANOVA. We also added a few sentences in the Results section describing the outcomes.

 3) The same applies to the statistical analyses in Figure 2. In addition, it will be easier to understand if Figure 2 is divided into 2 separate figures: one showing healthy samples, BPH and cancers, and the second comparing the 3 Gleason grades.

Response: The authors believe that, unlike ELISA, which is quantitative, the IHC staining scores are semi-quantitative.  Also, we cannot conduct ANOVA in Figure 2 because three of the six groups (groups 4-6) come from group 3 and are not independent.

 4) Can the authors provide better quality, higher contrast images for figures 3 and 4? The images are pale with a weak signal. In fact, even when zoomed in, the labeling cannot be clearly distinguished. It would be interesting to add an insert showing the image at higher magnification for each photo (3A 3B 3C 3D and 4A 4B 4C 4D).

Response: We have improved the quality and resolution of Figures 3 and 4 to 1200 dpi. We have included these improved Figures in the manuscript and provided a separate file containing these high-resolution Figures.

 5) There is still a long way to go to consider the H-score of GASP-1 applicable in routine. This is why a dedicated section on the limitations of the study would be beneficial in the discussion or conclusion. This must include the need for further validation in larger cohorts.

Response: We have added a separate paragraph at the end of the Discussion summarizing the study limitations as well as the need for further validation in larger cohorts

Reviewer 4 Report

Comments and Suggestions for Authors

The article of Cesar Torres-Luna and colleagues explores G protein coupled receptor associated sorting protein-1 (GASP)-1 as a more sensitive biomarker for prostate cancer (PCa) detection. The authors studied prostate tissue microarray of normal, benign prostatic hyperplasia (BPH) and prostate cancer patients with different Gleason. Immunoistochemistry and enzyme-linked immunosorbent assay (ELISA) analysis were performed using a polyclonal antibody against GASP-1. A 5-fold difference were found in the serum level of GASP-1 between BPH and PCa patients. The authors used a novel scoring system to assess GASP-1’ granules intensity and size and found a clear distinction between BPH and PCa patients. Moreover they found that GASP-1 overexpression correlates with PCa severity, providing insights into disease progression.

The authors conclude that GASP-1 can be a promising diagnostic marker, to be associated with PSA testing, and color analysis of granules may enhance diagnostic accuracy. Their GASP-1 IHC can be used to differentiate early and more advanced prostate cancer.

The study is interesting. GASP-1 IHC evaluation could be useful as biomarker to evaluate the presence of PCa, and its distinction from BPH, even if, as the authors say, the analysis of a larger number of patients sample would be required to validate their results. Further exploration in clinical settings is desiderable.

Some issues should be addressed:

Pag. 9, lines 292-295 . Why mini glands could be important if they contain fewer cancer cells, even if express more GASP-1?

Figures: Some time it is difficult to appreciate the expression of granules. The images are too blurry, pale

Author Response

We thank Reviewer 4 for the time to review our manuscript and for the positive feedback provided. Below are the comments from Reviewer 4 as well as our responses to those.

1. Pag. 9, lines 292-295 . Why mini glands could be important if they contain fewer cancer cells, even if express more GASP-1?

Response: Mini glands with a few cancer cells have been observed in early high-grade prostatic intraepithelial neoplasia (HG-PIN) tissue and the authors suggest that they may be associated with invasive phenotypes (Wang et al., J Cell Sci. 130: 104-110, 2017). To our knowledge, mini glands have not been observed in BPH. It is interesting to point out that the budding occurs from a specific subgroup of BPH that overexpresses GASP-1 in their basal cells (see Figure 3D). Because overexpression of GASP-1 is required for cancer progression and invasion, we speculate that such mini glands may promote BPH cell growth.

2. Figures: Some time it is difficult to appreciate the expression of granules. The images are too blurry, pale

Response: We have increased the resolution of Figures to 1200 dpi and the high-resolution figures have been added to the manuscript.  We have also provided a separate file containing these high-resolution Figures. In Figure 5, we also drew arrows to point out the granules.

Round 2

Reviewer 3 Report

Comments and Suggestions for Authors

Comments and Suggestions for Authors

2) Thank you for repeating the statistical analysis. However, I would appreciate it if you could adapt your graphical representation to suit statistical standards. The box-whisker plot is specific to non-parametric tests. Since ANOVA is a parametric test, the associated graph must be a bar graph.

4) The Image quality is much better. However, figures 5, 6, and 7 appear cropped on the right-hand edge. Please adjust the figure size to match the article format.

Author Response

We thank Reviewer 3 for the time to review our manuscript and for the positive feedback provided. Below are the comments from Reviewer 3 as well as our responses to those.

  1. Thank you for repeating the statistical analysis. However, I would appreciate it if you could adapt your graphical representation to suit statistical standards. The box-whisker plot is specific to non-parametric tests. Since ANOVA is a parametric test, the associated graph must be a bar graph.  Response: We have adapted the graphical representation and the graph is now a bar graph.
  2. The Image quality is much better. However, figures 5, 6, and 7 appear cropped on the right-hand edge. Please adjust the figure size to match the article format.  Response: Figures 5,6 and 7 have been adjusted and now they match the article format.